# Levels of TNF-α and Soluble TNF Receptors in Normal-Weight, Overweight and Obese Patients with Dilated Non-Ischemic Cardiomyopathy: Does Anti-TNF Therapy Still Have Potential to Be Used in Heart Failure Depending on BMI?

**DOI:** 10.3390/biomedicines10112959

**Published:** 2022-11-17

**Authors:** Elżbieta Lazar-Poloczek, Ewa Romuk, Wojciech Jacheć, Wiktoria Stanek, Bartosz Stanek, Monika Szołtysik, Tomasz Techmański, Maja Hasterok, Celina Wojciechowska

**Affiliations:** 1Second Department of Cardiology, Faculty of Medical Sciences in Zabrze, Medical University of Silesia, M. C. Skłodowskiej 10 Street, 41-800 Zabrze, Poland; 2Department of Biochemistry, Faculty of Medical Sciences in Zabrze, Medical University of Silesia, Jordana 19 Street, 41-808 Zabrze, Poland; 3Student Research Team at the Department of Biochemistry, Faculty of Medical Sciences in Zabrze, Medical University of Silesia, Jordana 19 Street, 41-800 Zabrze, Poland

**Keywords:** heart failure, tumor necrosis factor alpha, adipokines, obesity

## Abstract

Background. We sought to measure the levels of adipokines, TNF-α and soluble receptors (sTNFr1, sTNFr2) in heart failure patients with reduced ejection fraction (HFrEF) due to non-ischemic cardiomyopathy (nDCM). Methods. A total of 123 patients with HFrEF due to nDCM were divided into three groups according to BMI: 34 (27.6%) normal weight, 56 (45.5%) overweight and 33 (26.8%) obese. A six-minute walk test, echocardiography and right heart catheterization were performed. Serum concentrations of adiponectin, leptin, NT-proBNP, blood hemoglobin, sodium, creatinine, ALAT, AspAT, bilirubin, CRP, lipids, TNF-α, sTNFr1 and sTNFr2 receptors were measured. Results. Obese patients had the lowest NT-proBNP concentrations, significantly higher leptin levels and higher leptin/adiponectin ratios. The concentration of sTNFr1 was higher in normal-weight patients. In all groups, TNF-α concentrations correlated positively with sTNFr1 (*p* < 0.001). Higher levels of sTNFr1 were associated with higher sTNFr2 (*p* < 0.001) and CRP (*p* < 0.001). Moreover, the concentration of sTNFr2 positively correlated with CRP (*p* < 0.05) and adiponectin (*p* < 0.001). Levels of TNF-α were not associated with elevated CRP. Conclusion: This study demonstrated that changes in the concentrations of TNF and its receptors differ between groups of patients with different BMI. These findings suggest that the effective use of anti-TNF therapy is dependent not only on BMI, but also on concentrations of TNF-α receptors and other laboratory parameters.

## 1. Introduction

Heart failure is a clinical syndrome with different etiologies, both at the structural and cellular level. As the disease is still a common and potentially fatal condition, scientists are continually looking for new diagnostic methods and treatments to improve the quality of life and survival of patients. Last year, the new guidelines established a new recommendation for the treatment of chronic heart failure. The new, simplified treatment algorithm for the management of all HFrEF patients includes five groups of drugs [1]. In addition to the general recommendations, other therapies (e.g., ivabradine, hydralazine, angiotensin receptor blocker, cardiac resynchronization therapy) may be considered in selected patients according to their phenotypes. The guidelines also underlie the importance of non-cardiovascular comorbidities (e.g., diabetes, obesity, cachexia, anemia, inflammation) in heart failure. As overweight and obesity are considered risk factors for the development of heart failure, researchers have investigated adipose tissue as an endocrine organ with a great influence on the cardiovascular system. Moreover, despite the high incidence of cardiovascular disease in obese individuals, patients with heart failure and a higher body mass index (BMI) may survive for longer compared to their counterparts with a lower BMI [2,3,4,5].

Inflammation is known to play an important role in the pathogenesis of HF [6,7,8,9] and consequently may be an important therapeutic target for the treatment of HF [10,11]. Researchers have found that higher levels of pro-inflammatory cytokine tumor necrosis factor may lead to the progression of CHF, and there is a correlation between TNF-alpha (TNF-α) levels and the severity of HF [7,8], but, in fact, the role of TNF in the pathophysiology of HF remains unclear [12]. In addition to TNF, tumor necrosis factor receptors TNF r1 and TNF r2, responsible for the downstream cellular effects of TNF-α, are mainly assessed [13]. It is supposed that the activation of TNF r1 is deleterious, whereas the activation of TNF r2 is beneficial, and the ratio of the expression of these receptors in various tissue systems would potentially drive phenotypic variation [13,14,15].

These receptors, which play opposite roles in TNF-α signaling, can be released from various cells to generate soluble TNF receptors 1 or 2 (sTNF r1 or r2). Circulating TNF-α may bind to these receptors, which reduces the availability of TNF-α to bind and activate cell membrane receptors for TNF [16,17].

Although initial trials with high-dose anti-tumor necrosis factor alpha agents were associated with worsening HF [18,19], some recent studies do not show a detrimental effect of this therapy on cardiac function in rheumatoid arthritis patients [20]. Other studies reveal an improvement in ejection fraction or decrease in NT-proBNP during TNF blocking therapy [20,21]. In recent years, there has been renewed interest in TNF-alpha inhibitors as an anti-inflammatory therapy in HF. This is the reason that we consider the fact that therapy with TNF inhibitors is effective not in all patients but in a specific group of individuals [1].

This study sought to assess the relationships between serum TNF-α levels, sTNFr1 and sTNFr2 and other biomarkers and some clinical parameters in patients with HFrEF due to non-ischemic cardiomyopathy, stratified according to their BMI. Additionally, we examined the association between the TNF system and adipocytokines, released from the adipose tissue, such as leptin and adiponectin.

## 2. Materials and Methods

### 2.1. Study Group and Clinical Assessment

We enrolled 124 heart failure patients with a reduced ejection fraction due to non-ischemic cardiomyopathy. Patients were admitted to the 2nd Department of Cardiology, Zabrze, Poland, between 2008 and 2010, for periodic right heart catheterization in the management of heart transplant recipients. Dilated cardiomyopathy was diagnosed according to the WHO criteria. Patients were clinically stable, and their therapy had not been changed for at least one month before enrollment. Exclusion criteria were connective tissue disease, active inflammatory or infectious disorders, ischemic or valvular heart disease and alcohol abuse. All patients received optimal conventional HF therapy according to the current guidelines. The study protocol was approved by the Bioethics Committee of the Medical University of Silesia. Written informed consent was obtained from all patients.

### 2.2. Clinical Assessment

To assess functional capacity, the NYHA classification and six-minute walk test (6-minWT) were used. Body mass index was calculated by the following formula: weight in kilograms divided by the square of the height in meters (BMI). We performed a physical examination, ECG and echocardiography (VIVID 7, GE). Left ventricular end-diastolic volume (EDV) and end-systolic volume (ESV) were obtained from the apical four- and two-chamber views by the modified Simpson’s method. To determine ventricular systolic function, the left ventricular ejection fraction (LVEF) was calculated as follows: [EDV − ESV] × 100/EDV. Obstructive coronary artery disease was excluded by coronary angiography. Right heart catheterization (RHC) was performed with Swan-Ganz catheters (Star Edwards Lifesciences) under local anesthesia (1% Lignocaine) via the right jugular vein into the pulmonary artery. Pulmonary wedge pressure (PWP), systolic pulmonary artery pressure (sPAP), diastolic pulmonary artery pressure (dPAP) and right atrial pressure (RAP) were measured after a 20-min stabilization period. Systolic (sABP) and diastolic (dABP) arterial blood pressure were examined noninvasively. Cardiac output was measured by thermodilution with the use of a rapid bolus injection of 10 cc of cold saline. For the final evaluation, mean values of hemodynamic measurements (performed five times) were used [22].

### 2.3. Biochemical Methods

Blood samples for laboratory assessments were obtained from all patients on the day of clinical assessment at the start of the study. The samples were collected in the morning before breakfast, after 12 h of overnight fasting. The blood samples (5 mL) were centrifuged (10 min, 1500× *g*, 4 °C), and the serum was immediately separated and transferred into 1 mL cryotubes and stored at −70 °C for later analysis. The human adiponectin level was measured by sandwich enzyme-linked immunosorbent assay (ELISA) with a commercially available kit (Human Adiponectin ELISA, BioVendor, Karasek, Czech Republic). The concentration of adiponectin was expressed as μg/mL. The sensitivity of the assay was 26 ng/mL. Assay range was 0.1–10 μg/mL. Intra-assay was 4.9%, inter-assay 6.7%. The human leptin level was measured by sandwich enzyme-linked immunosorbent assay (ELISA) with a commercially available kit (Human Adiponectin ELISA, BioVendor, Karasek, Czech Republic). The concentration of leptin was expressed as ng/mL. The sensitivity of the assay was 0.2 ng/mL. Assay range was 1–50 ng/mL. Intra-assay was 5.9%, inter-assay was 5.6%. Human TNF-α, sTNFr1 (60 kDa) and sTNFr2 (80 kDa) levels were measured with a commercially available ELISA kit (BioVendor, Karasek, Czech Republic). The concentration of TNF-α was expressed as pg/mL. The sensitivity of the assay was 2.3 pg/mL. Assay range was 7.8–500 pg/mL. Intra-assay was 6.0%, inter-assay was 7.4%. The concentration of sTNFr1 (60 kDa) was expressed as ng/mL. The sensitivity of the assay was 0.05 ng/mL. Assay range was 0.08–5 ng/mL. Intra-assay was 1.9%, inter-assay was 8.6%. The concentration of sTNFr2 (80 kDa) was expressed as pg/mL. The sensitivity of the assay was 0.10 ng/mL. Assay range was 0.16–10 ng/mL. Intra-assay was 1.4%, inter-assay was 2.0%. All ELISA tests were performed using a BioTek Elx50 reader (BioTek Instruments Inc., Tecan Group, Männedorf, Switzerland).

Blood hemoglobin, sodium, creatinine, ALAT, AspAT, bilirubin, CRP and lipid parameters were determined using routine techniques on a Roche Cobas Integra 800. NT-proBNP was measured by the chemiluminescence method on a Roche Cobas 6000e501 (Roche Diagnostics GmbH, Mannheim, Germany).

### 2.4. Statistical Analysis

The study group was divided into three subgroups depending on their BMI. The normality of the distribution of the continuous data in the entire group of patients and each subgroup was analyzed using the Shapiro–Wilk test. Due to the nonparametric distribution, the continuous data were expressed as medians and first (Q1) and third (Q3) quartiles. Categorical data are presented as numbers and percentages. The significance of differences between groups was determined using the Kruskal–Wallis ANOVA test and, in the case of statistical significance, with the Chi2 with Yates correction and U Mann–Whitney tests as appropriate. The Spearman correlation coefficient was used to determine the correlations between biomarkers. Nonparametric tests were used due to the small sample size and non-normally distributed data. The results were considered statistically significant when *p* < 0.05. Statistical analyses were performed using Statistica 10.0 software (Statsoft, Inc., Tulsa, OK, USA).

## 3. Results

### 3.1. Clinical Characteristics

A total of 124 patients (18 females, 106 males), median age 47.8 [35.8–54.6] years, with HFrEF due to dilated cardiomyopathy were enrolled in the study. Only one female (0.1%) was considered underweight and cachectic with BMI = 17.3 and was not included in the analysis. In the entire group, 45.5% of patients were overweight (BMI: 25.00–29.99 kg/m^2^), 26.8% were obese (BMI: >30.00 kg/m^2^) and 27.6% were of normal weight (BMI: 18.00–24.99 kg/m^2^). Patients presented with severe left ventricular systolic dysfunction with median ejection fraction 25.0% (17.5–30.0%). Most patients (69.11%) were in NYHA functional class I or II. The clinical characteristics of the patients divided according to their BMI are summarized in Table 1. Surprisingly, the lowest percentage of patients in class I or II was observed among those with a normal body weight, and the best functional capacity in the 6-min walk test was observed in overweight patients. Echocardiographic and hemodynamic parameters were similar except for mean arterial pressure and cardiac output. The highest proportion of patients with hypertension and diabetes was observed in the obese group.

Most patients were given beta-blockers, angiotensin-converting enzyme inhibitors/angiotensin receptor blockers (ACE/ARB), aldosterone receptor antagonists (MRA), diuretics, digitalis or statins, but there was a small percentage of patients who reached target doses. The number of patients tolerating and treated with the maximum recommended doses of beta-blockers and ACE/ARB is provided in the second line of Table 1.

### 3.2. Laboratory Parameters

Table 2 compares basic laboratory parameters, inflammatory biomarkers, TNF, sTNFr1, sTNFr2, leptin and adiponectin and the ratio of TNF to receptors and leptin to adiponectin. The normal-weight patients showed lower sodium and increased bilirubin concentrations. Changes in TNF activity and adipokines are also depicted in Figure 1 and Figure 2. Laboratory tests in obese heart failure patients showed the lowest NT-proBNP concentrations, significantly higher leptin concentrations and higher leptin/adiponectin ratios. sTNFr1 levels were higher in normal-weight patients.

### 3.3. Correlation between TNF Activity, Adipokines and 6-minWT, NTproBNP and Clinical Parameters in the Entire Group of Patients

In the entire group, there were no statistically significant correlations between TNF activity and NT-proBNP, 6-minWT distance, echocardiographic and hemodynamic measurements.

### 3.4. Correlation between TNF Activity, Adipokines, CRP and BMI in the Entire Group of Patients

In the entire group, TNF-α concentrations correlated positively with sTNFr1 (r = 0.36, *p* < 0.001). Moreover, higher levels of sTNFr1 were associated with higher sTNFr2 (r = 0.35, *p* < 0.001) and CRP (r = 0.21, *p* < 0.05). In turn, the concentration of sTNFr2 positively correlated with adiponectin (r = 0.44, *p* < 0.001). Apart from the correlation with TNFR2, the concentration of adiponectin correlated negatively with the level of leptin (r = −0.39, *p* < 0.001). The leptin/adiponectin ratio correlated negatively with sTNFr2 (r = −0.20, *p* < 0.05). BMI correlated negatively with adiponectin (r = −0.20, *p* < 0.05) and positively with leptin (r = 0.28, *p* < 0.01) and leptin/adiponectin ratios (r = 0.30, *p* < 0.001)

### 3.5. Correlation between TNF Activity, Adipokines and 6-minWT, NT-proBNP and Clinical Parameters in Patients with HFrEF Stratified According to BMI

There was no correlation between TNF-α, TNF-α receptors and clinical echocardiographic and hemodynamic parameters and 6-minWT in any group. TNF-α, TNF-α/sTNFr1 and TNF-α/sTNFr2 correlated positively with NT-proBNP in overweight patients (r = 0.52, *p* < 0.01; r = 0.36, *p* < 0.05, respectively). In the normal-weight group, only sTNFr1 correlated positively and TNF-α/sTNFr1 correlated negatively with NT-proBNP (r = 0.42, *p* < 0.05; r = −0.57, *p* < 0.01). There was no correlation between TNF activity and NT-proBNP in obese individuals. In normal-weight patients, TNF-α positively correlated with sTNFr1 and sTNFr1, whereas sTNFr2 positively correlated with CRP; in turn, adiponectin correlated negatively with leptin concentrations. In obese patients, adiponectin positively correlated with TNF-α and negatively with leptin and with CRP (Table 3, Table 4 and Table 5). Correlations between TNF activity and NT-proBNP in patients categorized by their BMI are shown in Table 6.

## 4. Discussion

Various investigators have demonstrated that inflammatory mediators such as TNF-α play a role in the pathophysiology of chronic heart failure, and the role of TNF-α in the pathogenesis of heart failure is widely accepted [23,24,25,26]. Adipose tissue, apart from storing energy, is believed to play a protective role via soluble tumor necrosis factor alpha receptors, which can neutralize excessive TNF [27]. Therefore, the present study aimed to investigate the relationship between soluble TNF-α receptors—sTNFr1 and sTNFr2—and the severity of heart failure and inflammation to evaluate the potential use of TNF-α inhibitors in the management of heart failure.

The results of this study indicate that heart failure patients with a reduced ejection fraction due to non-ischemic dilated cardiomyopathy, stratified according to their BMI, despite similar echocardiographic and hemodynamic parameters, are likely to differ in functional capacity, comorbidities and some laboratory markers.

The most important finding is that TNF-α concentrations were similar in patients with normal body weight, overweight and obesity. Surprisingly, only sTNFr1 levels were higher in normal-weight patients. In the assessment of the relationship between TNF and its soluble receptors, the calculated ratio of TNF-α/sTNFr1 was highest in obese patients and lowest in overweight patients. The TNF-α/sTNFr2 ratio was comparable in all groups. The lack of differences in TNF-α levels between groups may be accounted for by the fact that an increase in TNF-α is associated not only with obesity, but also with cachexia [28].

Although the patient with BMI <18.5 kg/m^2^ was not included in the analysis, individuals with unintentional edema-free weight loss >5% in the last 12 months, suggesting cachexia, were not excluded. No association between BMI and TNF-α response was found in healthy volunteers subjected to experimental endotoxemia [29].

Zahorska et al. [30,31] demonstrated that TNF-α levels decreased after weight reduction, with an increase in both soluble TNF receptors. TNF-α binds rapidly to its soluble receptors, and, consequently, its concentration decreases suddenly or becomes even undetectable. Furthermore, there is evidence for the biological instability of plasma concentrations of TNF-α, in contrast to TNF receptors, which are characterized by easier detectability and a longer period of elevation in plasma. Therefore, measurements of TNF receptors appear to be more reliable. Some investigators have concluded that decreases in TNF-α levels and increases in soluble TNF receptors after slimming treatment may prevent the further loss of weight [30].

In our study, comorbidities (diabetes, arterial hypertension) were more common in obese patients. On the other hand, patients with a normal body weight were likely to have adverse prognostic factors, such as the lowest sodium concentrations, NYHA class III and IV and the highest NT-proBNP levels.

It is generally known that NT-proBNP is a useful biomarker in the diagnosis and prognosis of heart failure and is regarded as the gold standard in heart failure and cardiac dysfunction [32,33,34,35,36,37,38].

Recent evidence indicates also an inverse association between body mass index and NT-proBNP levels [39,40]. One of the postulated mechanisms for the decreased concentration of NT-proBNP in obese individuals is an imbalance in BNP receptors. The actions of BNP are mediated by two receptors: the natriuretic peptide receptor A (NPR-A), which activates the BNP pathways in different tissues, and the natriuretic peptide clearance receptor (NPR-C), which promotes BNP inactivation. Gentili et al. have demonstrated that obese patients have decreased levels of NPR-A and increased levels of NPR-C compared to lean patients [41].

Another possible explanation for the BNP degradation is the release of proinflammatory cytokines (IL-6, TNF) and resistin from the adipose tissue [42].

In the entire group, but not in overweight patients, there was a positive correlation between TNF-α and sTNFr1 levels.

The question remains as to whether the increase in sTNFr1 was compensatory and why it was not observed in overweight patients.

The findings of the present study show that elevated TNF-α was associated with elevated sTNFr1 in the entire group. After stratification into subgroups, this correlation was not observed in overweight patients. There was no relationship between TNF-α/sTNFr1 ratios and NTproBNP levels in the entire group or in obese patients. However, the relationship was negative in normal-weight patients and positive in overweight patients. Moreover, the concentration of TNF-α and the TNF-α/sTNFr2 ratio were associated with elevated NT-proBNP levels in overweight patients.

The question remains as to whether the increase in sTNFr1 is compensatory and why it is not observed in overweight patients.

Kleinborgard et al. found that sTNFr1, but not elevated sTNFr2, was associated with incident HF [43]. Several small cohort studies suggest that circulating levels of sTNFr1 and sTNFr2 are more strongly correlated with HF severity than TNF-α [44,45,46].

Marti et al. [47] confirmed the usefulness of sTNF-R1 in the prediction of incident HF in elderly patients, whereas elevated levels of TNF-α did not have additional predictive value compared with traditional HF risk factors. Considering the results of our study, we suggest that in normal-weight patients with HFrEF, the concentration of sTNFR1 increases; it may bind to TNF-α and inhibit its adverse effects (negative correlation with NT-proBNP). However, we did not observe that TNF-α promoted natriuretic peptide degradation in any group. Furthermore, BNP has the opposite effect to leptin, whose concentration in obese patients was elevated, in contrast to low values of NT-proBNP. Both mechanisms may stimulate aldosterone secretion, causing sodium and water retention. Packer et al. postulated that the leptin–aldosterone axis should be considered in the pathophysiology of heart failure and obesity [48].

Soluble TNF-α receptors seem to be more useful biomarkers because they are more stable than TNF in blood samples [30].

It is also worth mentioning that both sTNFr1 and sTNFr2, but not TNF-α, were associated with increased concentrations of CRP in normal-weight patients in the current study. Therefore, we propose to consider sTNF receptors rather than TNF-α as a biomarker of inflammation associated with heart failure.

In addition to analyzing the correlation between TNF and BMI, we assessed the relationship between TNF and adipokines produced by the adipose tissue. This study showed that obese patients had the highest leptin concentrations and leptin/adiponectin ratios. Similar findings were also reported by Wannammethe et al. In their study, leptin levels were significantly higher in overweight and obese patients with heart failure in comparison to subjects with a normal BMI. The investigators concluded that in men with HF, leptin (possibly reflecting cachexia) explained the inverse association between mortality and excess weight [49]. Further evidence for the presence of the obesity paradox in HF patients is the lack of correlation between leptin and TNFr1 harboring a death domain in obese patients with HF [50].

The current study found a negative correlation between leptin and sTNFr1 and a positive relationship between adiponectin and sTNFr2 in normal-weight patients. Hyperleptinemia in obesity is probably a result of the downregulation of leptin receptors, leading to leptin resistance. The role of leptin in heart failure is unclear. Some papers show that high concentrations of leptin may have a protective role in HF [51]. However, evidence also shows that leptin is associated with the progression of HF [52] and associated with poor prognosis [53]. The association between adiponectin and sTNFr2 was found in normal and overweight patients. Larger adipocytes in obese subjects produce lower levels of adiponectin but higher levels of proinflammatory cytokines, such as TNF-α [8]. Adiponectin inhibits the expression of TNF-α in adipocytes, and both TNF-α and IL-6 inhibit the production of adiponectin [10,11]. The negative regulation of adiponectin expression also results from hypoxia and oxidative stress [54,55]. Kistorp et al. reported an association between high adiponectin levels and increased risk of mortality in patients with chronic heart failure. Adiponectin levels were inversely correlated with BMI values. There was also an inverse correlation between leptin and adiponectin in each group. It was suggested that a high adiponectin level was related to increased energy expenditure, which might not be beneficial in patients with CHF [56].

In obese subjects, adiponectin levels are decreased, and the ability of adiponectin to inhibit the inflammatory processes is limited [57].

In our study, adiponectin was inversely related to CRP in obesity, but, surprisingly, leptin was inversely related to CRP in normal-weight patients. These unexpected results do not align with most previous observations [58,59]; however, a few investigators have noticed a similar correlation between leptin and CRP [60,61,62]. Consequently, this is an area for further research.

sTNFr1 and sTNFr2, but not TNF-α, were associated with increased CRP primarily in normal-weight patients. It is worth noting that patients in this group were least likely to receive statins with documented anti-inflammatory effects [63]. Obesity is a major risk factor for the development of heart failure [39,64,65,66,67,68,69]; therefore, it is worth considering which biomarkers should be used in obese patients. However, it is postulated that if patients already have heart failure, overweight and obese individuals are likely to have a surprisingly better prognosis compared to those with a normal body mass index [5,70,71,72,73,74,75,76,77]. This unexpected finding, commonly known as the “obesity paradox”, has been reported in several clinical cohorts. Numerous studies have now reported that the obesity paradox may be driven by lower cachexia, greater metabolic reserves, increased muscle mass and protective cytokines [77]. It is also likely that obese patients tolerate higher doses of cardioprotective drugs (due to higher blood pressure), and higher levels of lipoproteins in obese patients may combine with lipopolysaccharides and thus prevent the endotoxin-mediated proinflammatory cascade [78]. In the present cohort, obese patients received higher doses of ACE-I or ARB and statins.

In fact, recent reports have confirmed that a reduction in body mass index between the first and the second hospitalization for HF is associated with an increased risk for further hospitalizations and cardiovascular mortality [79].

This report suggests that changes in body weight reflecting cachexia may be more important than BMI. In line with these findings, a single assessment of BMI does not clearly identify patients with TNF disorders who could benefit from anti-TNF therapy.

On the other hand, the limitation of this hypothesis is the difficulty in the distinction between purposeful and non-purposeful weight loss [80,81]. Moreover, it is increasingly questionable whether the available epidemiological data show a better prognosis for obese patients. After eliminating other factors, such as smoking and co-morbidities including CVD, obese patients have a higher risk for mortality than people with a normal body weight [78].

Biological anti-TNF-α therapies are routine medications used in patients with rheumatoid and psoriatic arthritis, psoriasis, ankylosing spondylitis or Crohn’s disease. TNF-α inhibitors represent the fusion protein of TNF-α receptors linked to the Fc region of human antibody (etanercept) or chimeric (infliximab), fully human (adalimumab and golimumab) or modified human (certolizumab–pegol) anti-TNF-α antibodies [82]. One of the effects of this therapy is an increased BMI or body weight [82,83,84].

Based on the analyses performed, it was found that the behavior of TNF and its receptors differs depending on the BMI. However, based on our assessment, we do not believe that qualification for anti-TNF treatment can be based only on anthropometry (height, weight). The possibility of a change in body weight, the risk of sarcopenia or cachexia and evidence of inflammation should be taken into account. Moreover, the key to the benefit of this therapy may be determining the appropriate dose for patients with heart failure.

Therefore, we believe that future studies with a larger group of patients and with a more accurate assessment of cachexia should be conducted. Moreover, it is necessary to consider a study with patients treated according to the recent guidelines, including ARNI and SGLT-2 inhibitor therapy.

Undoubtedly, solutions to the above issues can be provided by studies that, on the one hand, assess the expression of TNF-α and its receptors, and, on the other hand, consider their influence on the cardiovascular system and long-term mortality in patients with heart failure.

## 5. Conclusions

In conclusion, laboratory tests showed that TNF-α concentrations were similar in patients with a normal body weight, overweight and obesity, whereas sTNFr1 levels were higher in normal-weight patients. There were no differences in the calculated ratios of TNF-α/sTNRr1, TNF-α/sTNFr2 and sTNFr1/sTNFr2 between the groups, either. No correlations between TNF activity and echocardiographic and hemodynamic parameters were found. The lowest NT-proBNP concentrations were detected in obese patients and the highest ones in lean patients. There were no correlations between NT-proBNP and TNF activity in obese patients.

This study showed that both sTNFr1 and sTNFr2, but not TNF-α, were associated with increased CRP only in normal-weight patients.

Thus, it is reasonable to assume that not only TNF-α concentrations but also the levels of its receptors are required to be measured before the implementation of anti-TNF therapy.

## 6. Study Limitations

We do, however, acknowledge that our study has certain limitations. First, the study group was relatively small. However, the group was homogenous, consisting of subjects with non-ischemic cardiomyopathy, and patients with the exacerbation of HF were not included in the study. Second, there was no information on the breakdown of muscle and fat tissues. Third, there were no data on weight changes prior to study entry.

## Figures and Tables

**Figure 1 biomedicines-10-02959-f001:**
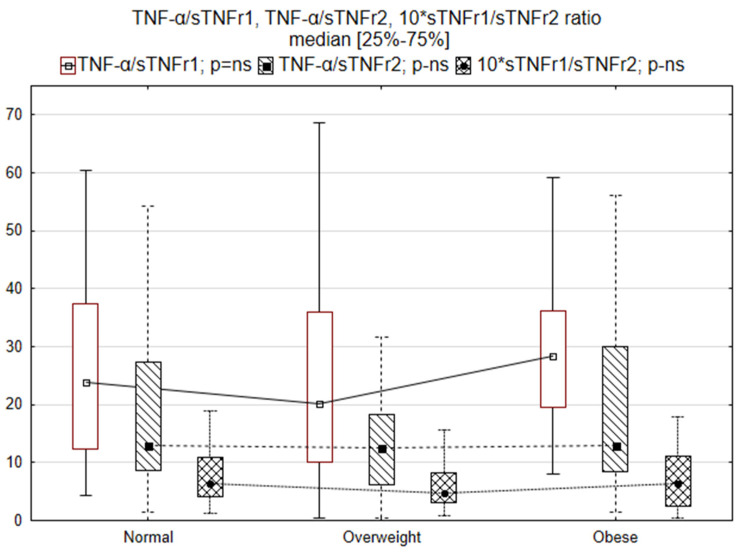
TNF-α/sTNFr1, TNF-α/sTNFr2 and sTNFr1/sTNFr2 ratios in normal-weight, overweight and obese patients.

**Figure 2 biomedicines-10-02959-f002:**
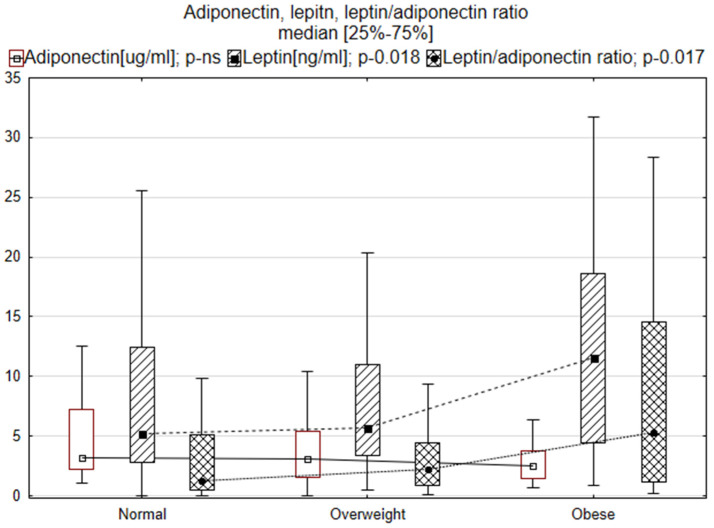
Adiponectin, leptin and leptin/adiponectin ratios in normal-weight, overweight and obese patients.

**Table 1 biomedicines-10-02959-t001:** The clinical characteristics of patients divided according to their BMI.

	Normal-Weight Patients (A) *n* = 34	Overweight Patients (B) *n* = 56	Obese Patients (C) *n* = 33	Kruskal–Wallis ANOVA Test
	Median [Q1–Q3] *n* [%]	Median [Q1–Q3] *n* [%]	Median [Q1–Q3] *n* [%]	
Age [years]	46.34 [33.40–53.90] *p* = 0.012 (vs. C)	41.45 [33.70–50.60] *p* < 0.001 (vs. C)	51.65 [46.50–56.65]	0.003
Sex F/M	6/28 [17.65/82.35]	8/48 [14.29/85.71]	3/30 [9.09/90.91]	0.827
NYHA I-II/III-IV	18/16 [52.94/47.06] *p* = 0.035 (vs. B)	43/13 [76.79/23.21]	24/9 [72.73/27.27]	0.011
6-minWT [m]	477.5 [413.0–548.0]	536.0 [464.0–580.0] *p* = 0.002 (vs. C)	453.0 [410.0–491.0]	0.006
b.w. [kg]	70.00 [65.00–75.00] *p* < 0.001 (vs. B & C)	84.00 [80.00–92.00] *p* < 0.001 (vs. C)	99.50 [90.00–105.0]	<0.001
AH	5 [14.70] *p* = 0.003 (vs. C)	3 (5.36) *p* < 0.003 (vs. C)	17 (51.52)	<0.001
Diabetes t2	1 [2.94] *p* = 0.007 (vs. C)	1 [1.79] *p* < 0.001 (vs. C)	10 [30.30]	<0.001
AF	6 [17.65]	13 [23.21]	6 [18.18]	0.493
EDD [mm]	68.50 [60.00–73.40]	68.00 [61.00–76.00]	69.00 [61.00–75.00]	0.827
EDV [ml]	195.0 [156.0–258.0]	206.0 [154.0–254.0]	200.0 [162.0–260.0]	0.917
LVEF [%]	20.00 [15.00–30.00] *p* = 0.018 (vs. C)	25.00 [15.10–30.00]	25.00 [20.00–31.00]	0.009
CO [ml/min]	4.00 [3.14–4.89] *p* = 0.011 (vs. C)	4.22 [3.75–5.17]	4.86 [3.83–5.67]	0.017
CI [ml/min/m2]	2.12 [1.77–2.61]	2.12 [1.93–2.45]	2.31 [1.94–2.59]	0.594
mean PAP [mmHg]	22.89 [16.33–37.57]	23.45 [19.67–36.33]	25.61 [17.93–36.48]	0.730
PCWP [mmHg]	16.70 [10.00–25.58]	18.00 [12.00–25.50]	17.67 [10.00–25.58]	0.752
HR [beat/min]	79.90 [65.92–88.50]	71.70 [64.00–84.00]	71.20 [66.20–76.90]	0.188
mean AP [mmHg]	84.85 [73.93–94.42] *p* = 0.011 (vs. C)	90.90 [82.67–100.0] *p* = 0.027 (vs. C)	97.17 [90.22–104.3]	0.002
Beta-blockers	33 [97.06] 7 [20.59]	54 [96.43] 17 [30.36]	30 [91.0] 10 [30.30]	0.622 0.557
ACEi/ ARB	32 [94.12] 4 [11.76] *p* = 0.078 (vs. B)	56 [100.0] 17 [30.36] *p* = 0.046 (vs. C)	33 [100.0] 15 [45.45] *p* = 0.020 (vs. A)	0.240 0.009
MRA	31 [91.17]	51 [91.07]	29 [87.88]	0.728
Diuretics	30 [88.24]	43 [76.79]	27 [81.82]	0.225
Digitalis	20 [58.82]	29 [51.78]	18 [54.55]	0.901
Statins	7 [20.59] *p* = 0.037 (vs. B)	25 [44.64]	21 [63.64] *p* < 0.001 (vs. A)	0.004
ICD	17 [51.50]	17 [30.36]	11 [33.33]	0.104
CRT	5 [14.71] *p* = 0.045 (vs. C)	10 [17.86] *p* = 0.025 (vs. C)	13 [39.39]	0.047

Q1÷Q3—first (Q1) and third (Q3) quartiles; F—female; M—male; BMI—body mass index; NYHA—New York Heart Association functional class; b.w.—body weight; AH—arterial hypertension; AF—atrial fibrillation; EDD—left ventricular end-diastolic diameter; EDV—end-diastolic volume; LVEF—left ventricular ejection fraction; 6-minWT—6-min walking test; CO—cardiac output; CI—cardiac index; PAP—pulmonary artery pressure; PCWP—pulmonary capillary wedge pressure; HR—heart rate; mean AP—systemic arterial pressure; ACEi—angiotensin convertase inhibitors; ARB—angiotensin receptor blockers; ICD—implantable cardioverter defibrillator; CRT—cardiac resynchronization therapy.

**Table 2 biomedicines-10-02959-t002:** Basic laboratory parameters and inflammatory biomarkers divided according to BMI.

	Normal-Weight Patients (A) *n* = 34	Overweight Patients (B) *n* = 56	Obese Patients (C) *n* = 33	Kruskal–Wallis ANOVA Test
	Median [Q1–Q3]	Median [Q1–Q3]	Median [Q1–Q3]	
Hemoglobin [g/dL]	13.7 [12.50–15.20]	14.4 [13.45–15.40]	14.30 [13.00–15.30]	0.294
NT-proBNP [pg/mL]	1590.0 [948.8–3000.0] *p* = 0.019 (vs. B)	861.00 [368.0–1692.0]	579.50 [259.5–1547.0]	0.002
Cholesterol [mg/dL]	164.0 [136.00–203.0]	183.00 [157.0–223.0]	189.50 [161.5–216.50]	0.130
HDL [mg/dL]	44.90 [29.00–51.00]	43.30 [36.05–54.85]	41.65 [35.90–45.00]	0.408
Triglycerides [mg/dL]	89.00 [75.00–123.0] *p* < 0.001 (vs. C)	118.5 [93.00–166.0] *p* = 0.012 (vs. C)	190.50 [117.5–262.0]	0.000
Bilirubin [µmol/L]	16.80 [11.30–38.90] *p* < 0.001 (vs. C)	14.40 [10.00–23.80]	12.15 [8.30–15.50]	0.003
Creatinine [µmol/L]	81.70 [64.20–96.20]	84.30 [75.80–93.90]	83.20 [74.20–90.80]	0.636
AspAT [IU/L]	25.00 [20.00–36.00]	23.00 [20.00–33.00]	25.00 [17.00–32.00]	0.766
AlAT [IU/L]	27.00 [17.00–0042.00]	28.00 [23.00–46.00]	30.00 [22.00–47.00]	0.879
Sodium [mmol/L]	135.50 [132.0–138.0] *p* = 0.002 (vs. C)	136.00 [134.0–139.00]	138.00 [136.00–140.0]	0.005
CRP [mg/L]	3.5 [2.49–5.03]	6.16 [2.46–9.23]	4.93 [2.59–8.36]	0.239
TNF-α [pg/mL]	8.54 [5.36–12.49]	7.05 [4.21–13.58]	9.11 [5.81–16.67]	0.379
sTNF r1 [ng/mL]	0.33 [0.21–0.47] *p* = 0.038 (vs. B)	0.25 [0.18–0.36]	0.25 [0.23–0.43]	0.012
sTNF r2 [ng/mL]	0.61 [0.35–0.89]	0.61 [0.30–0.89]	0.77 [0.30–0.98]	0.479
Adiponectin [µg/mL]	3.14 [2.24–7.24]	3.12 [1.54–5.42]	2.54 [1.41–3.76]	0.190
Leptin [ng/mL]	5.21 [2.80–12.45] *p* = 0.014 (vs. C)	5.71 [3.39–11.03] *p* = 0.021 (vs. C)	11.60 [4.47–18.61]	0.018

Q1÷Q3—first (Q1) and third (Q3) quartiles; NT-proBNP—N-terminal pro-B-type natriuretic peptide; HDL—high-density lipoprotein, AspAT—aspartate aminotransferase; AlAT—alanine aminotransferase; CRP—C-reactive protein; TNF-α—tumor necrosis factor α; sTNFr1—soluble tumor necrosis factor receptor-1; sTNFr2—soluble tumor necrosis factor receptor-2.

**Table 3 biomedicines-10-02959-t003:** Correlations between BMI, TNF activity and adipokines in normal-weight patients.

	TNF-α	sTNFr1	sTNFr2	Leptin	Adiponectin	CRP
TNF-α		r = 0.42 *p* < 0.05	ns	ns	ns	ns
sTNFr1	r = 0.42 *p* < 0.05		ns	r = −0.36 *p* < 0.05	ns	r = 0.37 *p* < 0.05
sTNFr2	ns	ns		ns	r = 0.63 *p* < 0.001	r = 0.50 *p* < 0.01
Leptin	ns	r = −0.36 *p* < 0.05	ns		r = −0.44 *p* < 0.01	r = −0.36 *p* < 0.05
Adiponectin	ns	ns	r = 0.63 *p* < 0.001	r = −0.44 *p* < 0.01		ns
CRP	ns	r = 0.37 *p* < 0.05	r = 0.50 *p* < 0.01	r = −0.36 *p* < 0.05	ns	

TNF-α—tumor necrosis factor α; sTNFr1—soluble tumor necrosis factor receptor-1; sTNFr2—soluble tumor necrosis factor receptor-2; CRP—C-reactive protein; ns—non significant.

**Table 4 biomedicines-10-02959-t004:** Correlations between BMI, TNF activity and adipokines in overweight patients.

	TNF-α	sTNFr1	sTNFr2	Leptin	Adiponectin	CRP
TNF-α		ns	ns	ns	r = −0.31 *p* < 0.05	ns
sTNFr1	ns		r = 0.51 *p* < 0.001	ns	ns	ns
sTNFr2	ns	r = 0.51 *p* < 0.001		ns	r = 0.50 *p* < 0.001	ns
Leptin	ns	ns	ns		*p* = −0.31 *p* < 0.05	ns
Adiponectin	r = −0.31 *p* < 0.05	ns	r = 0.50 *p* < 0.001	*p* = −0.31 *p* < 0.05		ns
CRP	ns	ns	ns	ns	ns	

TNF-α—tumor necrosis factor α; sTNFr1—soluble tumor necrosis factor receptor-1; sTNFr2—soluble tumor necrosis factor receptor-2; CRP—C-reactive protein, ns—non significant.

**Table 5 biomedicines-10-02959-t005:** Correlations between BMI, TNF activity and adipokines in obese patients.

	TNF-α	sTNFr1	sTNFr2	Leptin	Adiponectin	CRP
TNF-α		r = 0.39 *p* < 0.05	ns	ns	r = 0.34 *p* < 0.05	ns
sTNFr1	r = 0.39 *p* < 0.05		ns	ns	ns	ns
sTNFr2	ns	ns		ns	ns	ns
Leptin	ns	ns	ns		r = −0.33 *p* < 0.05	ns
Adiponectin	r = 0.34 *p* < 0.05	ns	ns	r = −0.33 *p* < 0.05		r = −0.42 *p* < 0.05
CRP	ns	ns	ns	ns	r = −0.42 *p* < 0.05	

TNF-α—tumor necrosis factor α; sTNFr1—soluble tumor necrosis factor receptor-1; sTNFr2—soluble tumor necrosis factor receptor-2; CRP—C-reactive protein, ns—non significant.

**Table 6 biomedicines-10-02959-t006:** Correlations between TNF activity and NT-proBNP in patients categorized by their BMI.

	NT-proBNP		
	Normal-Weight Patients (A)	Overweight Patients (B)	Obese Patients (C)
TNF-α	ns	r = 0.52, *p* < 0.01	ns
sTNF r1	r = 0.42, *p* < 0.05	ns	ns
sTNF r2	ns	ns	ns
TNF-α/sTNFR1 ratio	R = −0.57, *p* < 0.01	r = 0.36, *p* < 0.05	ns
TNF-α/sTNFR2 ratio	ns	r = 0.46, *p* < 0.05	ns
sTNFr1/sTNFr2 ratio	ns	ns	ns

TNF-α—tumor necrosis factor α; sTNFr1—soluble tumor necrosis factor receptor-1; sTNFr2—soluble tumor necrosis factor receptor-2, ns—non significant.

## Data Availability

The original data will be made available after contact with the corresponding author.

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
