# Peer review of "Levels of TNF-α and Soluble TNF Receptors in Normal-Weight, Overweight and Obese Patients with Dilated Non-Ischemic Cardiomyopathy: Does Anti-TNF Therapy Still Have Potential to Be Used in Heart Failure Depending on BMI?"

_biomedicines, 2022, doi:10.3390/biomedicines10112959_

Round 1

Reviewer 1 Report

 I have reviewed the manuscript by Lazar-Poloczek et al. entitled "Does anti-TNF therapy still have a chance of being used in heart failure?”

In this manuscript, the authors examined the concentrations of adipokines, TNF-alpha and soluble receptors (sTNFr1, sTNFr2) in patients with reduced ejection fraction heart failure (HFrEF) due to non-ischemic cardiomyopathy (nDCM), divided 3 groups according to BMI. In their study result, TNF alpha concentration was similar in patients with normal body weight, overweight and obesity. And the both sTNFr1 and sTNFr2, but not TNF-alpha were associated with increased inflammatory biomarkers CRP. 

In general, the possibility of anti-TNF therapy for heart failure is very important issues. However, in this manuscript they don’t answer the question and only presented the data.

The authors should either answer the question and mention the direction and plans for future clinical study on anti-TNF alpha therapy or change the title to something based on the authors’ results.

Reviewer 2 Report

Obesity is one of the most significant problems of the modern society, plays an important role in the occurrence of many diseases, in addition, aggravates their course. Modern therapy should take into account the obesity factor when recommending the treatment of cardiovascular diseases. However, this does not happen in all cases. This is partly due to the lack of knowledge about the changes in physiological and pathological processes depending on obesity. The presented manuscript explores the relationship between markers that characterize heart failure, including one of the important markers of cell death in heart failure TNF and its sTNFr1 and 2 receptors, and markers of metabolic disorders (leptin, adiponectin) and inflammation in patients with various degrees of obesity. The presented data are relevant and new. For the first time, it was found that in patients with obesity there is no relationship between sTNFr1 or 2 and an inflammatory marker. This is an important result that may influence clinical guidelines in the future. A good discussion of the results of the study is given.

Only one principal remark: The title does not reflect the results of the study.  A very pretentious name. It would fit a review article, but not an original study.

Minor remarks:

Abstract: The conclusion at the end of the abstract does not reflect the result of the study.

Introduction:

"some recent studies did not indicate the detrimental effect of this therapy on cardiac function in rheumatoid arthritis patients". You should add a link.

Results:

Figure 1: The data in Figure 1 partially repeats the data in Table 2. There is no indication of the significance of differences between groups. The same applies to figures 2 and 3. I see no reason to give figures 1-3, these data are presented in the table.

Correlations with NT-proBNP are given in the text, I advise you to bring them in a table or figure. This is an important result, it is better to present it visually.

Throughout the text - punctuation should be unified when citing in accordance with the rules of the journal ([12]. or .[16,17]).

Reasoning Notes

In patients with normal weight, bilirubin was elevated. Liver disease can be a source of sTNF. In addition, it is known that the maintenance of normal weight in patients with heart failure is often associated with smoking or alcoholism. Consider analyzing the sample along these dimensions.

Round 2

Reviewer 1 Report

Changing the title may be a good thing for readers.